# HPLC Analysis and the Antioxidant and Preventive Actions of *Opuntia stricta* Juice Extract against Hepato-Nephrotoxicity and Testicular Injury Induced by Cadmium Exposure

**DOI:** 10.3390/molecules27154972

**Published:** 2022-08-05

**Authors:** Xiaoli Zhu, Khaled Athmouni

**Affiliations:** 1College of Food Science & Institute of Food Biotechnology, South China Agricultural University, Guangzhou 510642, China; 2College of Food and Biotechnology, Guangdong Industry Polytechnic, Guangzhou 510300, China; 3Research Center for Micro-Ecological Agent Engineering and Technology of Guangdong Province, Guangzhou 510642, China; 4Laboratory of Animal Ecophysiology, Faculty of Sciences of Sfax, University of Sfax, Sfax 3000, Tunisia

**Keywords:** antioxidant activity, hepato-nephrotoxicity, HPLC analysis, *O. stricta*, testicular damage

## Abstract

*Opuntia stricta* is a rich source of phenolic compounds. This species generally has strong antioxidant activities in vitro and in vivo. This study aimed to analyze the antioxidant properties of phenolic compounds isolated from *Opuntia stricta*, including its radical scavenging activities and preventive action against Cd-induced oxidative stress in rats. To assess the protection of prickly pear juice extract (PPJE) against Cd-induced hepato-nephrotoxicity and testicular damage, male albino rats received PPJE (250 mg kg^−1^) and/or Cd (1 mg kg^−1^) by oral administration and injection, respectively, for five consecutive weeks. The preventive action of PPJE was estimated using biochemical markers of kidney and liver tissues, antioxidant status, and histological examinations. In the present study, the lipid peroxidation, protein carbonyls, antioxidant status, and metallothionein levels were determined in different tissues. The chromatographic analysis indicated that PPJE extract is very rich in phenolic compounds such as verbascoside, catechin hydrate, and oleuropein. Our results showed that PPJE-treated rats had significantly (*p* < 0.05) decreased Cd levels in liver and kidney tissues. In addition, the administration of PPJE induced a significant (*p* < 0.05) decrease in lipid peroxidation of 30.5, 54.54, and 40.8 in the liver, kidney, and testicle, respectively, and an increase in antioxidant status in these tissues. Additionally, PPJE showed a strong ability to protect renal, hepatic, and testicular architectures against Cd exposure. This study revealed that PPJE protects against the toxic effects of Cd, possibly through its free radical scavenging and antioxidant activities.

## 1. Introduction

Cadmium exposure can enhance oxidative stress in several organs, including kidney and hepatic tissues [1]. Heavy metals are known to induce carcinogenic toxicity [2] and adverse effects in humans, causing public health risks. However, they act as a threat to living organisms since they are highly toxic and accumulate in their body tissues [1]. Nevertheless, liver and kidney tissues perform detoxification and excretion actions. Among these metals, cadmium (Cd) is a very toxic metal that causes several alterations in human and animal tissues. Cd toxicity arises from its wide distribution in fertilizer, and consequently foods, and tobacco smoke [3] and is commonly used in industry development [4]. Usually, accumulation and intoxication phenomena occur in both liver and kidney tissues. Cd exposure can cause many alterations in kidney cells [2]. In addition, during Cd exposure, this metal accumulates predominantly in the liver, kidneys, reproductive organs, and tissues [5]. Principally, this heavy metal accumulation increases in both kidney and liver tissues [6]. Several studies have indicated that the mechanisms of toxicity of Cd exposure are: a reduction in glutathione levels (GSH), alterations in antioxidant enzymes, and enhanced ROS production in exposed tissues [7]. This unnatural level of ROS induces lipid peroxidation and oxidative DNA damage in cells [8]. Moreover, this metal causes several complications, such as liver and kidney injuries, respiratory diseases, neurological disorders, and testicular damage [9].

Medicinal plants generally have various preventive actions against several diseases [10]. In addition, the phenolic extracts of medicinal plants can reduce ROS levels in tissues [10]. Additionally, the preventive actions of natural antioxidant molecules against metal that cause several alterations are usually studied [8]. In recent years, studies on the antioxidant activities of medicinal plants have increased remarkably due to increased interest in their potential use as a rich and natural source of antioxidants. Many medicinal plants contain large amounts of antioxidants such as polyphenols, which can play an important role in absorbing and neutralizing free radicals, quenching singlet and triplet oxygen, or decomposing peroxides. The beneficial health effects of plants are attributed to flavonoids, a class of secondary metabolites that protect the plant against ultraviolet light and even herbivores.

In addition, cladode juice contains various compounds with high antioxidant potentials, including phenolic compounds, vitamins C and E, b-carotene (provitamin A), glutathione, etc. [11]. Moreover, several authors have reported that cladode juice exerts preventive actions against various alterations and enhanced Cd exposure [12]. This preventive action may be explained by the presence of the flavonoid compound quercetin 3-methyl ether in cladode extracts, which minimizes ROS levels in cells [13].

In the present study, the preventive action of prickly pear plant extract was evaluated against Cd toxicity. The prickly pear *Opuntia stricta*, a member of the Cactaceae family, is widespread in Mexico, much of Latin America, South Africa, and the Mediterranean area [14]. Accordingly, several authors have found that prickly pear has various pharmacological activities, including anti-proliferative and anti-viral [15], anti-inflammatory [16], and anti-hyperlipidemic properties [17] and analgesic action [18]. These data make prickly pear fruits and cladodes perfect candidates for cytoprotective investigations.

Interestingly, there are data in the literature showing that Opuntia extracts can be considered reliable and safe since no toxicity or only low toxicity has been found in animal models. An in vivo toxicity study suggests that the oral administration of *Opuntia ficus indica* extract at levels up to 2000 mg/kg/day does not cause adverse effects in male and female rats [19]. In an oral toxicity study, Sharma et al. [20] found that rats given the crude drug at doses up to 50 mL/kg exhibited no symptoms of toxicity. The crude extracts of *Opuntia* genus cladodes showed low toxicity in animal models [21,22].

The present study aimed to evaluate the hepato-nephro and testicular protective effects of the phenolic extract isolated from *O. stricta* cladode juice against Cd exposure by determining kidney, liver, and testicular biomarkers in male Wistar rats. 

## 2. Results

### 2.1. Phytochemical Determination and Antioxidant Potentials

The amounts of phenolic and flavonoid contents in PPJE are presented in Table 1. The juice extract of this Cactaceae showed high phenolic (24.71 ± 3.93 mg GAE/g DW) and flavonoid contents (8.84 ± 0.41 mg QE g^−1^ extract).

In the present work, the antioxidant activity of *O. stricta* was estimated by DPPH and ABTS methods. As shown in Figure 1, the values of DPPH radical scavenging activity varied between 20 and 83%. Concerning the ABTS radical scavenging capacity, the juice extract showed high antioxidant capacity (0.061 µM TE g^−1^ DW) (Table 1). 

### 2.2. HPLC Analysis of PPJE

The phenolic profile of PPJE is presented in Figure 2. Catechin hydrate, tyrosol, 4HOBenz, verbascoside, rutin, apigenin 7Glu, oleuropein, quercetin, pinoresinol and apigenin were found in *O. stricta*. Verbascoside was the major component (3.12 µg g^−1^), followed by catechin hydrate (1.43 µg g^−1^) and oleuropein (1.39 µg g^−1^). Minor phenolic compounds included pinoresinol, quercetin, and apigenin (Table 2).

### 2.3. Effects of PPJE Cd-Induced Damage in Rats

#### 2.3.1. Markers of Hepatic and Nephrotoxicity Toxicity

Cd administration significantly decreased (by 40%) the levels of ALT, AST, and bilirubin in the liver compared with the control group after 5 weeks (Table 3). The Cd-treated rats had significantly (*p* < 0.05) enhanced plasma levels of creatinine and urea in kidney tissue as compared to the normal rats. Treatments of rats with PPJE decreased the levels of AST, ALT, and bilirubin in the liver and creatinine and urea in the kidney compared with the Cd group.

#### 2.3.2. Enzymatic Antioxidants

The levels of the antioxidant enzymes SOD, CAT, and GPx in the liver, kidney, and testicular tissues of the experimental animals are given in Table 4, Table 5 and Table 6. Cd treatment led to a significant decrease in hepatic SOD, CAT, and GPx (−48.44%, −48.29%, and −6.9%, respectively) compared to those of the control group. After five weeks of Cd exposure, levels of enzymatic activity such as SOD, CAT, and GPx in kidney and testicular tissues were significantly increased compared to the normal group (*p* < 0.05). Our results indicated that PPJE-treated rats were able to protect the antioxidant status from Cd toxicity. 

#### 2.3.3. Lipid Peroxidation and Protein Oxidation Indices

Cd exposure induced increases in lipid peroxidation (LPO) of 30.5%, 54.54%, and 40.8% in the liver, kidney, and testicle, respectively, compared with the Cd-treated group (Table 4, Table 5 and Table 6). In addition, the protein carbonyl contents in the liver, kidney, and testicle were significantly higher in the Cd-treated group compared with control rats, as shown in Table 4, Table 5 and Table 6, respectively. In addition, the administration of ethanolic PPJE (250 mg kg^−1^) alone to rats caused a significant reduction in LPO and protein carbonyl levels in these different organs. 

#### 2.3.4. Effects of Cd Exposure on MT Concentration in Rat Liver and Kidney

Metallothionein is widely considered a sensitive marker of oxidative stress. The metallothionein concentrations in the liver and kidney tissues were enhanced in the Cd-treated group compared with untreated rats (Figure 3a,b). Additionally, the present study indicated that the treatments of ethanolic PPJE (250 mg kg^−1^) significantly decreased the MT concentration in both organs when compared with Cd-treated rats.

#### 2.3.5. Cadmium Estimation

The Cd concentrations in hepatic and kidney tissues are given in Table 7. The results of the ANOVA test showed that Cd content in these organs was significantly higher (*p* < 0.05) than that in normal rats. The amount of Cd content varied among different groups and ranged from 0.01 to 0.46 µg g^−1^ w.t.w in the liver, whereas this metal varied between 0.01 and 1.34 µg g^−1^ w.t.w in kidney tissue. The highest Cd concentration was measured in the Cd group. In contrast, the administration of PPJE extract (250 mg kg^−1^) decreased the concentrations of Cd in liver and kidney tissues when compared to the Cd-treated group.

### 2.4. Effects on Histopathological Changes

The histopathological changes in the liver and kidneys are presented in Figure 4 and Figure 5, respectively. The liver and kidneys of controls showed normal morphologies (Figure 4a and Figure 5a, respectively). In contrast, treatment with Cd alone produced focal hepatocyte swelling, vacuolation and inflammation (leukocyte infiltration), focal proximal tubule degeneration (Figure 4c), and glomerular swelling (Figure 5c) in the kidneys. Administration of PPJE (250 mg kg^−1^) preserved the morphology and also restored the architectures of the liver and kidney tissues (Figure 4d and Figure 5d, respectively).

After 5 weeks of Cd exposure, the histopathological examination of testicular tissues showed testicular alterations comprising edematous vasculitis and stromal hemorrhage. Many seminiferous tubules were edematous and undergoing degeneration. Spermatogenesis was almost absent (Figure 6c). In the control and juice extract alone groups, the histology was similar (Figure 6a,b). In rats co-treated with Cd and juice extract, a significant reduction in the restoration of spermatogenesis in most of the seminiferous tubules was observed (Figure 6d).

## 3. Discussion

Polyphenol compounds have attracted considerable attention because of their various biological activities, including: antioxidant, antimutagenic, antitumor, and anti-inflammatory activities [23]. Our results showed that the PPJE extract exhibited a high content of phenolic compounds and flavonoids. This is similar to the results reported by another study [12]. Concerning the qualitative analysis of the phenolic extract, major phenolic compounds were isolated and identified from many Tunisian medicinal plants, such as *O. stricta.* The major types and representative components of natural compounds (catechin hydrate, tyrosol, 4-hydroxybenzoic acid (4HOBenz), verbascoside, rutin, apigenin 7glucoside (apigenin 7Glu), oleuropein, quercetin, pinoresinol, and apigenin) were found in *Opuntia stricta*. Verbascoside was the major component, followed by catechin hydrate and oleuropein. Minor phenolic compounds included pinoresinol, quercetin, and apigenin. Several studies have found that these natural compounds have anti-inflammatory, analgesic, and antioxidant effects [12,24].

Extracts of active compounds from natural plants provide potent protection to the biological system against the damaging effect of natural oxidation processes in the organism. In this investigation, the antioxidant capacity of PPJE extract was evaluated by two assays. Each antioxidant assay possesses its own unique mechanism to evaluate the antioxidant activity in the sample. Our results in this study showed that ethanolic PPJE exhibited high antioxidant capacity. It is worth noting that the extract produced with ethanol presented high antioxidant potential and also a high content of total phenols.

The present study is the first to investigate and reveal the protective impact of juice extract on liver and kidney metabolism under Cd exposure. Our results indicated a significant increase in serum ALT, AST, unconjugated bilirubin, creatinine, and urea after Cd administration to rats. The highest levels of these enzymes in plasma represent biomarkers of hepato-nephrotoxicity [25]. Moreover, Cd exposure significantly elevated serum hepatic and kidney marker enzymes [26]. The highest levels of these enzymes are a marker of cell damage [26]. Indeed, these radicals adversely affect the antioxidant system of the organism [27]. The important enzymes of this system are SOD, CAT, and GPx, which protect cells against reactive oxygen species (ROS). Cd toxicity is associated with the elevation of ROS levels, DNA damage, and lipid peroxidation in vitro. In the current study, the levels of oxidative stress in these three organs were significantly decreased compared to levels in the control group, which indicated that Cd was able to induce serious oxidative stress. Cd induces the accumulation of superoxide anion in cells, which is why we studied the effects of this metal on SOD activity. Concerning these enzymes, the inhibitory action of Cd on SOD may be due to competition between Cd and Zn or Cu (cofactors of SOD activity) [28]. In addition, this heavy metal altered the transport systems of calcium (Ca), Fe, Zn, Cu, and Mg [29]. These elements represent cofactors of the antioxidant system. In addition, in vivo administration of Cd altered the SOD activity. Nguyen et al. [30] found that Cd exposure induced the subcellular accumulation of hydrogen peroxide. Indeed, a higher H_2_O_2_ concentration might be implicated in the induction of catalase activity. The reduction in catalase activity in this investigation may be explained by the Cd-catalyzed oxidation of peroxisomal proteins, inducing carbonylation, particularly of the CAT enzyme [31]. In the present investigation, the decrease in antioxidant status due to Cd was accompanied by an elevation of hepatic protein carbonyls. The decrease in catalase activity by Cd may be explained by a decrease in iron absorption, an essential trace element required for CAT activity [32]. The effect of Cd exposure on glutathione peroxidase (GPx), which plays an important role in the detoxification of xenobiotics, was studied in the liver and kidney of Wistar rats. Our results indicate that the Cd-induced decrease in glutathione peroxidase activity may arise as a consequence of Se-mediated detoxification of Cd, where the Se level is insufficient to maintain optimal GPx activity [33]. Moreover, the administration of PPJE ameliorated the SOD, CAT, and GPx activities in Cd-treated rats. Accordingly, Eneman et al. [34] indicated that phenolic compounds were able to modulate the transcription and expression of proteins related to antioxidant enzymes. In addition, Cd exposure induced the peroxidation of membrane lipids of cells in various organs by stimulating reactive oxygen species. These free radicals bind to cellular macromolecules and stimulate lipid peroxidation and protein oxidation. In the present work, the marker of lipid peroxidation and protein destruction (protein carbonyl contents) decreased in the PPJE-treated group compared to the control. Several studies found that hepato-nephrotoxicity induced by Cd may be prevented by antioxidant supplementation, which are present in medicinal plants [35]. In general, polyphenols are known to be able to protect cell membrane integrity, protecting cells from death. Phenolic compounds are reported to be potent antioxidants and protect tissues from the toxic effects of Cd exposure [36]. The effect of Cd was also detected on the metallothionein (MT) levels. The present findings indicated that MT levels significantly increased in Cd-treated rats. This protein has been implicated in the scavenging of heavy metals by forming trimercaptide linkages [37]. The binding of Cd to MT is considered a mechanism of cell defense as MT sequesters, transports, and inactivates metal ions. The administration of juice extract decreased the concentration of MT in liver and kidney tissues compared with untreated rats. This diminution of Cd-induced alteration in Mt expression is connected with the ability of these compounds to chelate Cd [38].

## 4. Materials and Methods

### 4.1. Preparation and Extraction of Opuntia Stricta Cladode Powder

Fresh cladodes from *Opuntia stricta* were collected from the area of Sidi Bouzid (Tunisia) (latitude 35°2′25″ N, longitude 9°29′37″ E; elevation: 41 m) throughout February 2015. Experts of the Plant Biology Department of the University of Sfax confirmed plant identity. Voucher specimens were deposited in the National Gene Bank of Tunisia. After sample preparation, cladode juice (500 mL) was extracted using ethanol (75%) at room temperature for 48 h. Finally, the homogenate was condensed under reduced pressure by a rotary evaporator, and the yield of this extract was calculated.

### 4.2. Phytochemical Properties of O. stricta Juice Extract

#### 4.2.1. Determination of Total Phenolic Content

The total phenols of cladode juice of this plant sample were determined using Folin–Ciocalteu’s phenol reagent [39]. The results were determined at 765 nm using a colorimetric assay. The total phenolic content was expressed as mg gallic acid equivalent (GAE) mg^−1^ of dry weight. All analyses were performed in triplicate.

#### 4.2.2. Determination of Total Flavonoids

Aluminum chloride (AlCl_3_) was used to evaluate the flavonoid content in *O. stricta* [40]. Catechin was used as a standard. The results were expressed as mg catechin equivalents g^−1^ of dry weight. All analyses were performed in triplicate.

### 4.3. Antioxidant Properties of O. stricta

#### 4.3.1. Diphenyl−2-Picrylhydrazyl (DPPH) Radical Scavenging Activity

The DPPH radical scavenging activity of the samples was determined using the method described by Ozturk et al. [41]. The optical density was measured spectrophotometrically at 515 nm. The percent of inhibition (PI) was measured according to the following formula:Inhibition (%) = [(A_control_ − A_test_)/A_control_] × 100(1)
where A_control_ is the absorbance of the control, and A_test_ is the absorbance of the juice extract. All samples were measured in triplicate.

#### 4.3.2. Free Radical Scavenging Ability with the Use of ABTS Radical Cation (ABTS Assay)

The antioxidant potential of the cladode juice extract of *O. stricta* (PPJE) was also evaluated by determining their capacity to minimize the ABTS^•+^ free radical using the method reported by Ozgen et al. [42]. The final result was expressed as µM of Trolox equivalents (TE) per g of dry weight.

### 4.4. High-Performance Liquid Chromatography Analysis (HPLC) of PPJE

The PPJE samples were subjected to HPLC analysis using a Varian Prostar HPLC equipped with a C 18 reverse phase column (Varian, 250 mm × 4.6 mm, particle size 5 µm), a ternary pump (model Prostar 230), and a Prostar 330 diode array detector with gradient elution. Phenolic compounds in the sample were quantified using standard curves of standard solutions injected into the HPLC. The flow rate was 1 mL min^−1^, and the injection volume was 20 µL at 30 °C. The identifications were performed at 290 nm for phenolic acids and at 365 nm for flavonoids based on the comparison with the retention times of standards and by co-injection. The quantification of these compounds was carried out by comparing the areas of the peaks with an internal standard (resorcinol). The result was expressed as µg of phenols g^−1^ of dry weight.

### 4.5. Experimental Design

Albino male Wistar rats aged 3–4 months, weighing 180 ± 20 g, and purchased from the Central Pharmacy (SIPHAT, Ben Arous, Tunisia) were used in the present study. They were housed at room temperature (37 °C) in a light/dark cycle of 12 h and a relative humidity below 40%. They had free access to a commercial pellet diet (SNA, Sfax, Tunisia) and tap water. The Committee of Animal Ethics of Sfax approved the experimental protocols. In our experiment, we used 24 rats. The Tunisian ethics committee for the care and use of laboratory animals approved the handling of the animals. Three weeks after acclimation to laboratory conditions, rats were randomly divided into four groups, each with six rats.

Group 1 (control group) received normal saline for 5 weeks.

Group 2 (Opuntia stricta only) received cladode juice extract (250 mg kg^−1^ body wt/day) orally by gavage for 5 weeks.

Group 3 (cadmium chloride (CdCl_2_)-treated group) received CdCl_2_ (1 mg kg^−1^ body wt/day.p.i.) for 5 weeks.

Group 4 (cadmium chloride/Opuntia stricta juice co-administration) received CdCl_2_ (1 mg kg^−1^ body wt/day) orally by gavage concurrently with cladode juice extract (250 mg kg^−1^ body wt day^−1^) for 5 weeks.

Five weeks later, the rats were sacrificed by decapitation, and their trunk blood was collected in EDTA tubes. The serum was prepared by centrifugation (3500× *g*, 15 min, 4 °C). Other blood samples were immediately used for the determination of serum enzymes and other biochemical indices.

All samples were stored at −80 °C until used. For histological studies, sections of the liver, kidney, and testicle were stored in 4% formalin solution. Sections of 5 μm thickness were stained with hematoxylin–eosin.

### 4.6. Biochemical Biomarker Assays

The serum urea, creatinine, alanine, and aspartate aminotransferase activities (ALT and AST) were measured using commercial kits (from Biolabo, Maizy, France) on an automatic biochemistry analyzer (Vitalab Flexor E, Irvine, CA, USA).

### 4.7. Enzymatic Antioxidant Status

Liver, kidney, and testicle homogenates were used for the evaluation of enzymatic status: superoxide dismutase was determined by the method of Beauchamp and Fridovich [43], catalase was measured as described by Aebi [44], and glutathione peroxidase (GPx) was determined using the method developed by Flohé and Günzler [45].

### 4.8. Oxidative Stress Biomarkers

Lipid peroxidation was estimated calorimetrically by measuring thiobarbituric acid-reactive substances (TBARS), as developed by Niehaus [46]. The protein oxidation level was detected in the liver and kidney by determining the total protein carbonyl content using a technique developed by Levine [47] and expressed as nmol/mg protein.

### 4.9. Determination of MT Concentration

Metallothionein (MT) levels in liver and kidney tissues were evaluated using Ellman’s reagent [0.4 mM 5,5′-dithiobis-(2-nitrobenzoic acid) (DTNB) in 100 mM KH_2_PO4] at pH 8.5. This reagent was mixed with NaCl (2 M) and 1 mM EDTA. Then, aliquots of homogenate of each organ were homogenized in three volumes of 0.5 M sucrose and 20 mM Tris–HCl buffer (pH 8.6) with the addition of 0.006 mM leupeptin, 0.5 mM phenyl methylsulfonyl fluoride (PMSF), and 0.01% 2-mercaptoethanol. The homogenate was then centrifuged at 15,000× *g* for 30 min at 4 °C. The obtained supernatant was treated with ethanol/chloroform as described by Viarengo et al. [48] in order to obtain the MT-enriched pellet. The MT pellet was resuspended in HCl/EDTA in order to remove metal ions still bound to the MT. Finally, the interaction between thiol and DTNB reagent was detected in NaCl solution (2 M).

### 4.10. Cadmium Estimation

After acid digestion, the Cd content in the liver and kidney was estimated by atomic absorption spectrometry (Perkin-Elmer, model: 370, Waltham, MA, USA). The results are expressed as micrograms of Cd per gram of wet tissue weight (µg/g w.t.w).

### 4.11. Histopathological Studies

Sections of the kidney, liver, and testicle were fixed in 4% formalin solution and embedded in paraffin. Finally, tissue preparations were examined and photographed using an Olympus CX41 microscope (Tokyo, Japan).

### 4.12. Statistical Analysis

Mean values of different assays were used in variance (ANOVA) analysis with IBM SPSS Statistics version 20. The significance level was determined (*p* < 0.05), and significant differences were measured according to Duncan’s Multiple Range Test (DMRT) with a confidence level of 95%.

## 5. Conclusions

In summary, these findings clearly show that *O. stricta* juice extract exhibited a high amount of phenolic and flavonoid contents, as well as antioxidant capacity. The juice extract of *O. stricta* could protect the hepatic and kidney tissues of rats from Cd-induced oxidative damage. The faithful mechanisms of protection offered by the investigated juice extract may involve the scavenging of free radicals generated during Cd metabolism in vivo and/or the induction of antioxidative enzymes. In addition, juice extract containing certain phenolic compounds has a strong ability to scavenge free radicals and stimulate the antioxidant system of rats. Moreover, studies are required to explain the detailed molecular mechanism of protection of this juice extract against Cd-induced toxicity.

## Figures and Tables

**Figure 1 molecules-27-04972-f001:**
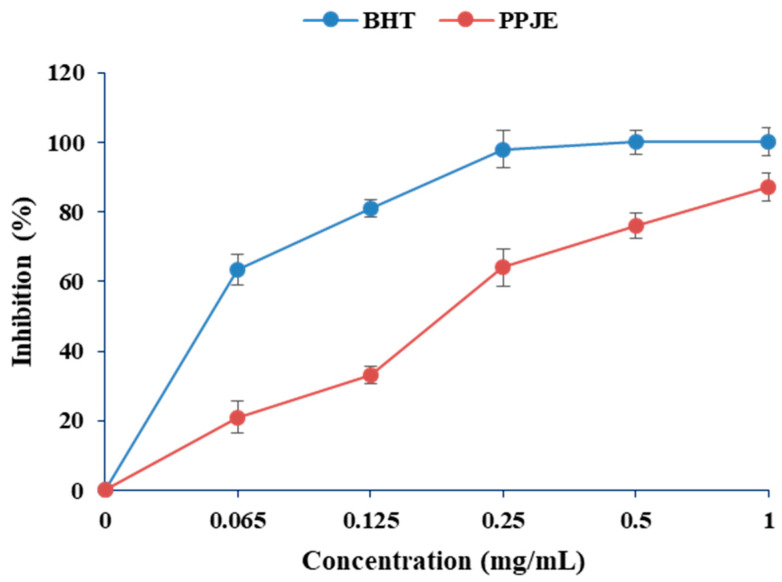
Effects of extracts of *O. stricta* on *in vitro* free radicals (DPPH).

**Figure 2 molecules-27-04972-f002:**
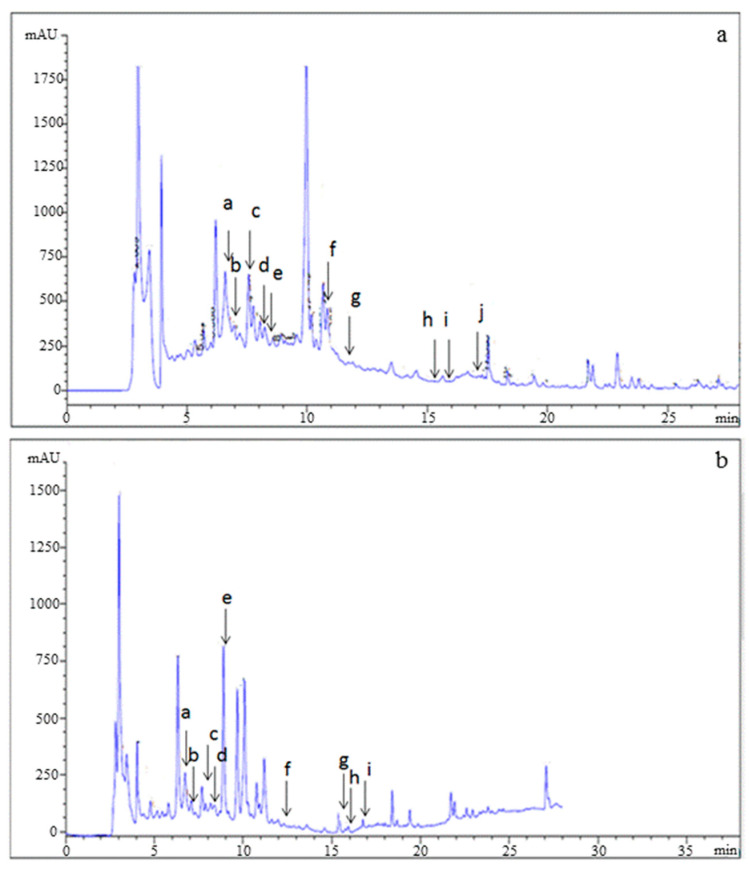
HPLC analysis of (**a**) standards and (**b**) juice extract of *O. stricta* cladode.

**Figure 3 molecules-27-04972-f003:**
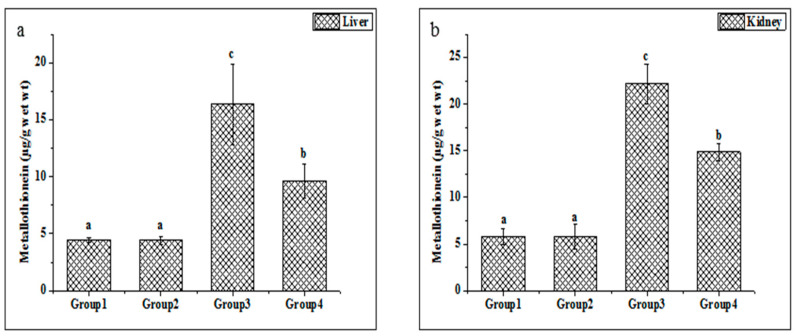
Effect of juice extract of *O. stricta* on Cd-induced changes in the levels of metallothionein in the liver (**a**) and kidney (**b**) of control and experimental rats. Group 1: Normal rats. Group 2: Juice extract (250 mg/kg)-treated rats. Group 3: CdCl_2_ (1 mg/kg)-treated rats. Group 4: CdCl_2_ + juice extract (250 mg/kg)-treated rats. Values are mean ± SD for 6 rats in each group. Bars not sharing a common superscript letter (a, b, c) differ significantly at *p* < 0.05 (Duncan).

**Figure 4 molecules-27-04972-f004:**
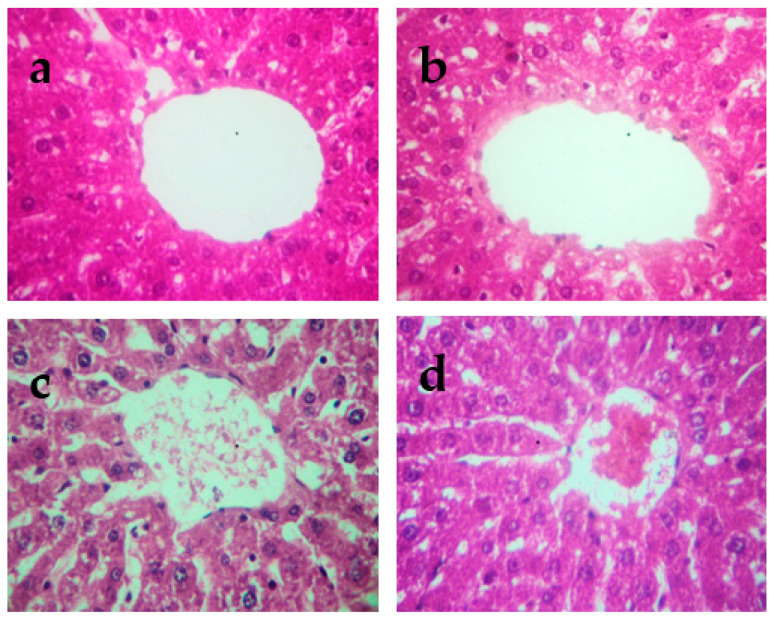
Representative photographs from the liver showing the protective effect of juice extract against Cd-induced hepatic damage in rats (H&E 40×). (**a**) Normal rat liver showing normal hepatic parenchyma and intact central vein. (**b**) Juice extract (250 mg/kg)-treated rat liver showing normal appearance of hepatocytes around the central vein. (**c**) CdCl_2_ (1 mg/kg)-treated rat liver showing extensive degeneration of hepatocytes with focal necrosis, vacuolated cytoplasm, inflammatory cell infiltration, and damaged central vein. (**d**) CdCl_2_ + juice extract (250 mg/kg)-treated rat liver showing near-normal hepatic architecture and normal histological features.

**Figure 5 molecules-27-04972-f005:**
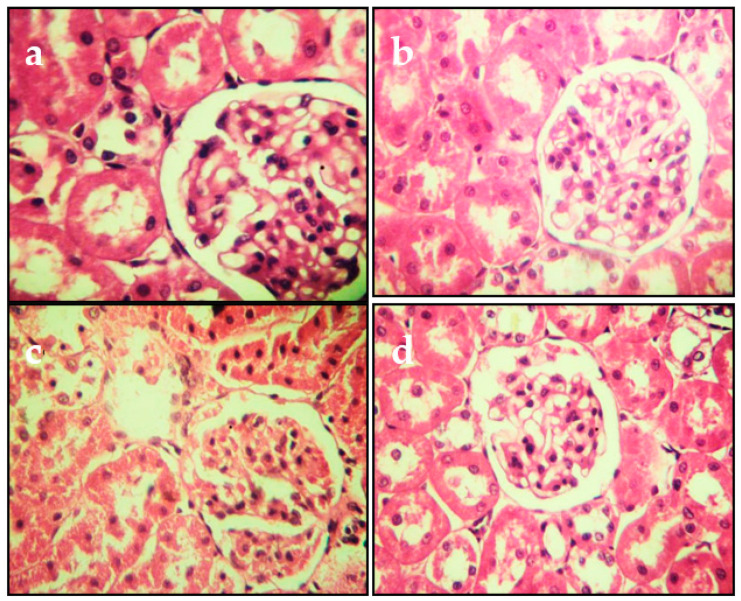
Representative photomicrographs of section from kidney. (**a**) Control group. (**b**) Juice extract (250 mg/kg)-treated rat kidney showing normal appearance of glomeruli. (**c**) CdCl_2_ (1 mg/kg)-treated rat kidney showing tubule glomerular degeneration. (**d**) CdCl_2_ + juice extract (250 mg/kg)-treated rat kidney showing near-normal kidney architecture and normal histological features. All sections were stained with hematoxylin/eosin; 400× for all panels.

**Figure 6 molecules-27-04972-f006:**
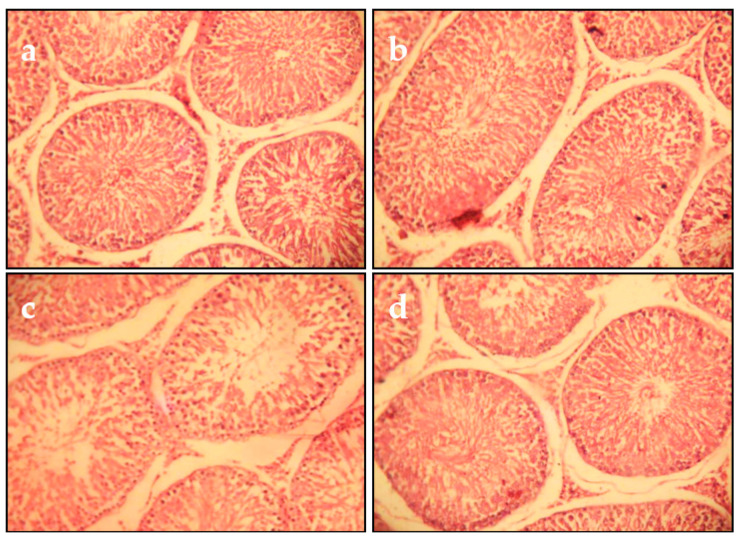
Microscopic evaluation of testicular tissue from juice extract of *O. stricta* alone, Cd and juice co-treated groups at 5 weeks (H&E 400): (**a**) Control group. (**b**) Juice extract (250 mg/kg)-treated group showing that the seminiferous tubular cells and interstitial tissue were normal. (**c**) Section of testes from rats treated with Cd (1 mg/kg, for 5 weeks); interstitial tissues showed edema, hemorrhage, and vacuolation, and seminiferous tubules were edematous with intact germinal layer and undergoing degeneration along with loss of spermatogenesis. (**d**) Section of testes from rats treated with Cd (5 weeks) and juice extract of *O. stricta* (250 mg/kg).

**Table 1 molecules-27-04972-t001:** Total phenolic content and total flavonoids of juice extract of *O. stricta* cladode.

	Juice Extract of *O. stricta* Cladode
Total phenol (mg GAE/g DW)	24.71 ± 3.93
Flavonoids (mg QE/g DW)	8.48 ± 0.43
ABTS (µM TE/g DW)	0.061 ± 0.001

Data expressed as average ± SD (*n* = 3) standard deviations. GAE: gallic acid equivalent; QE: quercetin equivalent; TE: Trolox equivalent; DW: dry weight.

**Table 2 molecules-27-04972-t002:** Chemical composition of ethanol extracts from *O. stricta* cladode juice by HPLC analysis.

Short Name	Retention Time (min)	Composition (µg/g)
Catechin hydrate	6.71	1.43
Tyrosol	7.06	1.10
4-Hydroxybenzoic acid	7.67	1.34
Verbascoside	8.88	3.12
Rutin	8.15	1.29
Apigenin 7glucoside	10.75	1.10
Oleuropein	11.95	1.39
Quercetin	15.40	0.21
Pinoresinol	15.90	0.11
Apigenin	16.74	0.27
Luteolin-7-Glu	Nd	Nd

Nd: not detected.

**Table 3 molecules-27-04972-t003:** Serum ALT, AST, bilirubin, urea, and creatinine of the studied groups. Control group. Juice extract (250 mg/kg)-treated rat kidney showing normal appearance of glomeruli. CdCl_2_ (1 mg/kg)-treated rat kidney showing tubule glomerular degeneration. CdCl_2_ + juice extract (250 mg/kg)-treated rat.

Group	ALT	AST	Bilirubin (µmol/L)			Creatinine (mmol/L)	Urea (mmol/L)
	(IU/L)	(IU/L)	Total	Conjugated	Unconjugated		
Control	22.31 ± 2.31 ^a^	112.42 ± 9.65 ^a^	0.43 ± 0.08 ^a^	0.41 ± 0.07 ^a^	0.02 ± 0.00 ^a^	8.51 ± 0.54 ^a^	38.65 ± 4.65 ^a^
PPJE extract	21.54 ± 1.46 ^a^	114.54 ± 7.64 ^a^	0.38 ± 0.09 ^a^	0.36 ± 0.02 ^a^	0.02 ± 0.00 ^a^	11.31 ± 1.32 ^a^	41.32 ± 7.34 ^a^
CdCl_2_	43.54 ± 2.14 ^c^	135.76 ± 5.74 ^c^	1.46 ± 0.21 ^c^	1.14 ± 0.06 ^b^	0.32 ± 0.04 ^c^	17.64 ± 2.21 ^b^	65.81 ± 5.67 ^c^
CdCl_2_ + PPJE extract	28.64 ± 1.65 ^b^	123.54 ± 5.82 ^b^	1.08 ± 0.13 ^b^	0.94 ± 0.07 ^b^	0.14 ± 0.06 ^b^	11.32 ± 1.57 ^a^	51.64 ± 7.07 ^b^

^a–c^ Values having different letters on the same line showed significant differences (*p* < 0.05).

**Table 4 molecules-27-04972-t004:** Antioxidant enzyme activities and stress biomarkers levels in liver tissue of the studied groups. Control rats. PPJE extract (250 mg/kg)-treated rats. CdCl_2_ (1 mg/kg)-treated rats. CdCl_2_ + PPJE extract (250 mg/kg)-treated rats. α: U mg^−1^ protein; β: µmoles/H_2_O_2_ consumed min^−1^ mg^−1^ of protein; γ: nmoles GSH min^−1^ mg^−1^ of protein; δ: nmoles of MDA g^−1^ of tissue; ε: nmoles mg^−1^ of protein.

	Control	PPJE Extract	CdCl_2_	CdCl_2_ + PPJE Extract
SOD ^α^	28.28 ± 3.41 ^d^	27.37 ± 1.74 ^cd^	14.58 ± 2.54 ^a^	23.07 ± 2.04 ^b^
CAT ^β^	37.64 ± 2.45 ^d^	36.85 ± 1.75 ^c^	19.46 ± 1.82 ^a^	31.25 ± 3.05 ^b^
GPx ^γ^	362.33 ± 7.84 ^d^	361.05 ± 7.42 ^c^	337.32 ± 9.64 ^a^	353.61 ± 6.54 ^b^
LPO ^δ^	0.74 ± 0.11 ^b^	0.72 ± 0.08 ^a^	1.34 ± 0.42 ^d^	0.93 ± 0.21 ^c^
Protein carbonyl ^ε^	1.81 ± 0.13 ^b^	1.78 ± 0.21 ^a^	4.37 ± 0.37 ^d^	2.36 ± 0.17 ^c^

^a–d^—Values having different letters on the same line showed significant differences (*p* < 0.05).

**Table 5 molecules-27-04972-t005:** Antioxidant enzyme activities and stress biomarkers levels in kidney tissue of the studied groups. Control rats. PPJE extract (250 mg/kg)-treated rats. CdCl_2_ (1 mg/kg)-treated rats. CdCl_2_ + PPJE extract (250 mg/kg)-treated rats. α: U mg^−1^ protein; β: µmoles/H_2_O_2_ consumed min^−1^ mg^−1^ of protein; γ: nmoles GSH min^−1^ mg^−1^ of protein; δ: nmoles of MDA g^−1^ of tissue; ε: nmoles mg^−1^ of protein.

	Control	PPJE Extract	CdCl_2_	CdCl_2_ + PPJE Extract
SOD ^α^	41.44 ± 1.47 ^c^	42.24 ± 2.45 ^c^	18.07 ± 1.65 ^a^	34.51 ± 1.84 ^b^
CAT ^β^	51.31 ± 3.14 ^c^	52.62 ± 1.84 ^c^	28.21 ± 2.74 ^a^	41.21 ± 1.87 ^b^
GPx ^γ^	375.31 ± 3.24 ^d^	373.34 ± 5.62 ^c^	342.27 ± 6.04 ^a^	361.07 ± 7.86 ^b^
LPO ^δ^	0.71 ± 0.12 ^b^	0.68 ± 0.08 ^a^	2.31 ± 0.32 ^d^	1.05 ± 0.11 ^c^
Protein carbonyl ^ε^	1.85 ± 0.23 ^b^	1.77 ± 0.32 ^a^	6.34 ± 0.63 ^d^	2.74 ± 0.14 ^c^

^a–d^—Values having different letters on the same line showed significant differences (*p* < 0.05).

**Table 6 molecules-27-04972-t006:** Antioxidant enzyme activities and stress biomarkers levels in testicular tissue of the studied groups. Control rats. PPJE extract (250 mg/kg)-treated rats. CdCl_2_ (1 mg/kg)-treated rats. CdCl_2_ + PPJE extract (250 mg/kg)-treated rats. α: U mg^−1^ protein; β: µmoles/H_2_O_2_ consumed min^−1^ mg^−1^ of protein; γ: nmoles GSH min^−1^ mg^−1^ of protein; δ: nmoles of MDA g^−1^ of tissue; ε: nmoles mg^−1^ of protein.

	Control	PPJE Extract	CdCl_2_	CdCl_2_ + PPJE Extract
SOD ^α^	38.51 ± 2.07 ^c^	37.65 ± 1.67 ^c^	15.32 ± 0.84 ^a^	27.84 ± 2.23 ^b^
CAT ^β^	47.21 ± 2.54 ^c^	48.74 ± 1.67 ^c^	24.82 ± 3.75 ^a^	38.74 ± 2.34 ^b^
GPx ^γ^	365.31 ± 2.31 ^c^	364.43 ± 4.08 ^c^	338.72 ± 5.72 ^a^	357.63 ± 3.65 ^b^
LPO ^δ^	0.65 ± 0.42 ^a^	0.66 ± 0.11 ^a^	2.23 ± 0.22 ^c^	1.32 ± 0.08 ^b^
Protein carbonyl ^ε^	1.64 ± 0.34 ^a^	1.63 ± 0.21 ^a^	5.54 ± 0.37 ^c^	2.86 ± 0.42 ^b^

^a–c^—Values having different letters on the same line showed significant differences (*p* < 0.05).

**Table 7 molecules-27-04972-t007:** Effect of *O. stricta* juice extract on Cd content in liver and kidney tissues. Control rats. PPJE extract (250 mg/kg)-treated rats. CdCl_2_ (1 mg/kg)-treated rats. CdCl_2_ + PPJE extract (250 mg/kg)-treated rats.

Groups	Cd Concentration µg g^−1^ Dry Mass	
	Liver	Kidney
Control	0.01 ± 0.00 ^a^	0.01 ± 0.00 ^a^
PPJE extract	0.01 ± 0.00 ^a^	0.01 ± 0.01 ^a^
CdCl_2_	0.46 ± 0.05 ^c^	1.34 ± 0.08 ^c^
CdCl_2_ + PPJE extract	0.24 ± 0.03 ^b^	0.97 ± 0.07 ^b^

^a–c^—Values having different letters on the same line showed significant differences (*p* < 0.05).

## Data Availability

The authors declare that all data supporting the findings of this study are available within the paper.

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
