# Peer review of "HPLC Analysis and the Antioxidant and Preventive Actions of *Opuntia stricta* Juice Extract against Hepato-Nephrotoxicity and Testicular Injury Induced by Cadmium Exposure"

_molecules, 2022, doi:10.3390/molecules27154972_

Round 1

Reviewer 1 Report

The study explored the phenolic compounds from Opuntia stricta their protective effects against Cd-induced cell damage by evaluating their antioxidant activity and evaluation of biochemical marker levels in kidney and liver tissues, DNA damage, antioxidant status, and pathohistological examinations of affected animal organs. The paper is overall technically sound and merits publication. However, the authors are suggested to be guided with the ff comments

  1. The authors must carefully check for format and grammatical errors. There are many misspelled words and there are floating figures!
  2. What is the relevance of assessing fatty acid profiles? The paper is thought to be focused only on phenolics.
  3. The authors must present the tables involving animal studies by explaining in each table the basis of groupings. While this was explained in the methodology, they must be reindicated as footnotes in the results for clarification and to avoid confusion.

Author Response

Reviewer#1 comments

Question 1: The authors must carefully check for format and grammatical errors. There are many misspelled words and there are floating figures!

Response 1: We agree with the reviewer’s assessment. These errors have been corrected thoroughly.

Question2: What is the relevance of assessing fatty acid profiles? The paper is thought to be focused only on phenolics.

Response 2: We agree with the reviewer’s assessment. The fatty acid profiles have been removed from this manuscript.

Question 3: The authors must present the tables involving animal studies by explaining in each table the basis of groupings. While this was explained in the methodology, they must be reindicated as footnotes in the results for clarification and to avoid confusion.

Response 3: This sentence has been checked.

Reviewer 2 Report

The manuscript presents an interesting topic regarding study of role of Opuntia stricta extract in preventing cadmium-caused physiological disorders in rats. The idea of the research was generally good. However, the submitted manuscript needs significant improvement before publication. Specific comments are provided below:

  • Line 16, 89: The methodology states that it was collected (Line 306) and the other plece in manuscript states that it was cultured in vitro (Opuntia stricta in vitro ) please be more specific and agree on one version.
  • From which plant fragment were the bioactive compounds extracted? Please elaborate on the description
  • Please specify in the abstract the changes in which tissues, and give numerical values for the most important results (e.g., by how much did the lipid peroxidation, antioxidant status)
  • Line 28: This sentence in unclear, please rewritten.
  • Line 35-53: Information is not consistent, please rewritten.
  • Line 59: Please changes phenolic molecules for phenolic compounds
  • Line 60, 67: Please changes auteurs for authors
  • Line 72-74: Please specify whether the purpose was to study the effect of plant extract or juice extract. It is incorrect to use them interchangeably.
  • Line 271, 333, 326, 392, 394, 398, 400, 403: Please correct to by Name et al. [year].
  • Line 315: Please add cladode juice extract
  • Line 359: This subsection is too genral, please be more specific
  • Line 405: Please correct were evaluated using using for evaluated using
  • Line 423: What confidence levels were assigned to each analysis, please annotate and verify throughout the manuscript.
  • Line 105: Please correct apigeninwere for apigenin were
  • Line 109: please correct ND* for Nd-
  • Line 116, 115: Please put a space after the superscript
  • Line 123, 136: A different significance value is given in the statistical description, please correct
  • The Unit is repeated twice in Table 4
  • Line 127: Please changes groupes for treatment and groupe 1,2,3,4 for group 1,2,3,4
  • Line 138, 151, 156: In Table 5,6,7 please adopt a different designation for SOD, CAT, GPx, LPO, Proteine carbonyl, they may not be the same as statistical significance. Please changes ‘’Values having different letters on a same line showed significant difference (P<0.05)’’ for ‘’ a-d- Values having different letters on a same line showed significant difference (P<0.05)’’.
  • Line 144: Please detail in which tissues and by how much it reduced LPO
  • Line 167: Please correct prouvd for provide
  • Line 285: Please add numer refrence
  • Line 222: Please add refrence in the sentece: ‘’It is similar to results reported by another research’’
  • Line 239-250: Compared to which other extraction, the ethanolic extract tested was most effective in terms of antioxidant activity? Please add the missing information
  • Line 251: Please rewritten this sentence: ‘’The toxicity test was determined after the in vitro evaluation of ethanolic PPJE.’’

Author Response

Dear Editor,

Thank you for your email dated consideration of our manuscript entitled “HPLC analysis, antioxidant and the potential role of Opuntia stricta extract in preventing Cadmium-caused physiological disorders in rats: biochemical and molecular evidence". We wish to thank the editor and reviewers for their helpful comments. We have now thoroughly revised the manuscript by incorporating all changes suggested. The corrections are indicated by red colour in the revised version.

Reviewer#2 comments

Question 1: Line 16, 89: The methodology states that it was collected (Line 306) and the other plece in manuscript states that it was cultured in vitro (Opuntia stricta in vitro ) please be more specific and agree on one version.

Response 1: We agree with the reviewer’s assessment. These errors have been corrected thoroughly.

Question2: From which plant fragment were the bioactive compounds extracted? Please elaborate on the description

Response 2: Phenolic extract has been extracted from Opuntia stricta cladode.

Question 3: Please specify in the abstract the changes in which tissues, and give numerical values for the most important results (e.g., by how much did the lipid peroxidation, antioxidant status)

Response 3: This sentence has been checked.

Question 4: Line 28: This sentence is unclear, please rewritten.

Response 4: Correction is made.

Question 5: Line 35-53: Information is not consistent, please rewritten.

Response 5: In response to the reviewers’ suggestion. This sentence has been checked.

Question 6: Please changes phenolic molecules for phenolic compounds

Response 6: We agree with the reviewer’s assessment. The mistake has been corrected thoroughly.

Question 7: Line 60, 67: Please changes auteurs for authors.

Response 7: This sentence has been revised as you suggested.

Question 8: Line 60, 67: Please changes auteurs for authors

Response 8: This mistake has been corrected.

Question 9: Please specify whether the purpose was to study the effect of plant extract or juice extract. It is incorrect to use them interchangeably.

Response 9: This mistake has been corrected.

Question 10: Line 271, 333, 326, 392, 394, 398, 400, 403: Please correct to by Name et al. [year].

Response 10: All corrections are made. 

Question 11: Line 315: Please add cladode juice extract

 Response 11: This correction is made

Question 12:   Line 359: This subsection is too genral, please be more specific

Response 12: This sentence has been revised.

Question 13:  Line 105: Please correct apigeninwere for apigenin were

Response 13: Correction is made.

Question 13:  Line 105: Please correct apigeninwere for apigenin were

Response 13: Correction is made.

Question 14:  Line 109: please correct ND* for Nd-

Response 14: Correction is made.

Question 15:  Line 116, 115: Please put a space after the superscript

Response 15: Correction is made.

Question 16:  Line 123, 136: A different significance value is given in the statistical description, please correct. The Unit is repeated twice in Table 4

Response 16: Correction is made.

Question 17:  Line 127: Please changes groupes for treatment and groupe 1,2,3,4 for group 1,2,3,4

Response 17: Correction is made.

Question 18:  Line 138, 151, 156: In Table 5,6,7 please adopt a different designation for SOD, CAT, GPx, LPO, Proteine carbonyl, they may not be the same as statistical significance. Please changes ‘’Values having different letters on a same line showed significant difference (P<0.05)’’ for ‘’ a-d- Values having different letters on a same line showed significant difference (P<0.05)’’.

Response 18: Correction is made.

Question 19:  Line 144: Please detail in which tissues and by how much it reduced LPO

Response 19: Correction is made.

Question 20:  Line 167: Please correct prouvd for provide

Response 20: Correction is made.

Question 21:  Line 285: Please add numer refrence

Response 21: Correction is made.

Question 22:  Line 222: Please add refrence in the sentece: ‘’It is similar to results reported by another research’’

Response 22: Correction is made.

Question 23:  Line 239-250: Compared to which other extraction, the ethanolic extract tested was most effective in terms of antioxidant activity? Please add the missing information.

Response 23: Correction is made.

Question 24:  Line 251: Please rewritten this sentence: ‘’The toxicity test was determined after the in vitro evaluation of ethanolic PPJE.’’

Response 24: Correction is made.

Reviewer 3 Report

Comments to authors

The current paper explores the potential of Opuntia stricta extract in preventing Cadmium-induced damage in multiple organs before identifying chemical constituents in the extract using GC/MS and HPLC.

Specific comments:

Title: Authors are advised to improve the title – what is “physiological disorders”? This is too broad.

Introduction: There are multiple redundant sentences in the section. The structure needs to be improved significantly as currently it is rather difficult to follow through.

Results:

Cadmium is written in full or as sometimes as Cd throughout the manuscript – please check for consistencies.

All tables: Please cross check the consistencies in decimal separator.

Table 4 – 8: Please indicate what each group is. 

Figure 5: Missing ladder. Exposure might be too high? Why is the lower right of the image white?

Figure 6: Please indicate what each group is.

Figure 7, 8, 9: Please improve the histological studies. There are some images showing different hue/saturation and it is difficult to make comparison between groups. Missing scale bar.

What are the additional images under Figure 8 (which got cropped)?

Methods:

Different animal models are used for the calculation of LD50 and toxicity analysis. Could the authors provide justification on this (rat vs mice)?

Please indicate ethics approval number for the current study.

How long were the animals were administered with cadmium/extract? Please provide more details.

Any positive controls were included for toxicity studies?

Discussion:

The authors are encouraged to revise the entire discussion section.

Line 221: “ethanolic PPJE exhibited the highest content of phenolic compounds and flavonoids content.” – comparison with which extract? Only one extract is studied here?

Line 236: “Several studies…antioxidant effects.” – One citation only?

Line 251: What is DL50?

Cadmium is written in full or as sometimes as Cd throughout the manuscript – please check for consistencies.

The image for gel electrophoresis remains the same (white patch and no ladder marker) and it's best to have ladder marker as (positive) control and "ruler" to estimate size of the nucleic acids for the experiment.  

Table 3: "Groups" not "groupes"

Table 4: Extra space for "Group 4"

It would be good to rename all the groups for tables/figures (e.g. Control, Opuntia stricta extract, CdCl2-treated model, CdCl2-treated + Opuntia stricta extract).  

Author Response

Dear Editor,

Thank you for your email dated consideration of our manuscript entitled “HPLC analysis, antioxidant and the potential role of Opuntia stricta extract in preventing Cadmium-caused physiological disorders in rats: biochemical and molecular evidence". We wish to thank the editor and reviewers for their helpful comments. We have now thoroughly revised the manuscript by incorporating all changes suggested. The corrections are indicated by red colour in the revised version.

Reviewer#3 comments

Question 1: Title: Authors are advised to improve the title – what is “physiological disorders”? This is too broad.

Response 1: We agree with the reviewer’s assessment. The title has been changed

Question2: Introduction: There are multiple redundant sentences in the section. The structure needs to be improved significantly as currently it is rather difficult to follow through..

Response 2: We agree with the reviewer’s assessment. The sentence has been revised.

Question 3: Cadmium is written in full or as sometimes as Cd throughout the manuscript – please check for consistencies.

Response 3: Correction is made

Question 4: All tables: Please cross check the consistencies in decimal separator.

Response 4: Correction is made.

Question 1: Table 4 – 8: Please indicate what each group is.

Response 1: In response to the reviewers’ suggestion. The mistake has been checked.

Question 1: Figure 5: Missing ladder. Exposure might be too high? Why is the lower right of the image white?

Response 1: We agree with the reviewer’s assessment. It’s due to the light of the camera

Question 2: Figure 6: Please indicate what each group is.

Response 2: Correction is made.

Question 3: Please improve the histological studies. There are some images showing different hue/saturation and it is difficult to make comparison between groups. Missing scale bar.

Response 3: We agree with the reviewer’s assessment. I only have these images and to do the study again requires a long time.

Question 5: What are the additional images under Figure 8 (which got cropped)?

Response 5: this is a mistake when converting the word form to PDf form

Question 6: Different animal models are used for the calculation of LD50 and toxicity analysis. Could the authors provide justification on this (rat vs mice)?

Response 6: We agree with the reviewer’s assessment. We always use mice to determine the toxicity of substances  

Question 7: Any positive controls were included for toxicity studies?.

Response 7: In this study no.

Question 8:   Line 221: “ethanolic PPJE exhibited the highest content of phenolic compounds and flavonoids content.” – comparison with which extract? Only one extract is studied here?

Response 8: This sentence has been corrected.

Question 9:  Line 236: “Several studies…antioxidant effects.” – One citation only?

Response 9: Correction is made.

Question 10:  Line 251: What is DL50?

Response10: median lethal dose (LD50), or median lethal concentration (LC50), is a quantitative indicator of the toxicity of a substance. This concept also applies to irradiations. This indicator measures the dose of the substance causing the death of 50% of a given animal population (often mice or rats) under specific experimental conditions. This is the mass of substance needed to kill 50% of the animals in a batch. It is expressed in milligrams of active ingredient per kilogram of the animal. We determine it to choose the right dose of our extract

Question 11:  Cadmium is written in full or as sometimes as Cd throughout the manuscript – please check for consistencies.

Response 11: Correction is made.

Round 2

Reviewer 2 Report

The submitted manuscript entitled ''HPLC analysis, antioxidant, and the preventive action of Opuntia stricta juice extract against hepato-nephrotoxicity, testicular injury, and DNA damage induced by cadmium exposure'', despite minor revisions by the authors, still requires further significant revisions. The suggested changes are outlined below. The authors did not follow the Instructions for Authors (''All Figures, Schemes, and Tables should be inserted into the main text close to their first citation and must be numbered following their number of appearance (Figure 1, Scheme I, Figure 2, Scheme II, Table 1, etc. ''), placing tables and figures at the end of the manuscript. Moreover, the figures and tables presented are illegible. In addition, the authors did not address all the comments, or in some cases despite the comment: ''Correction is made'' the changes were not made in the manuscript.

•  Fig 1. Should be corrected. The methodology does not indicate that this is an in vitro method but a basic laboratory analysis.

•             In the abstract, statistical significance is reported at the 0.01 level, while in the statistical description, the confidence level was set at 0.05. Please verify this information.

• ''[26] found that Cd-exposure induced subcellular accumulation of hydrogen peroxide'', Please correct to by Name et al. [year].

• In many cases, spaces are missing between words/characters in the manuscript.

• In many cases, ''et al.'' is missing. This should be verified with the references and corrections made.

•             The subsection ''Enzymatic antioxidant status'' is not numbered and is not in italics

• The subject of this study was not cladode juice but cladode juice ethanol extract, please detail this information in the missing places.

• Please correct were evaluated ''using using'' for evaluated using

• Conducting appropriate statistical analysis is critical to the research being conducted. What confidence levels in ''4.11. Statistical analysis'' were assigned to each analysis, please annotate and verify throughout the manuscript.

• In Table 4,5,6, the abbreviation denoted in the superscript (''SOD a, CAT b, GPx c, LPO d, Proteine carbonyl e'') may not be the same as the denotation of statistical differences between the results presented (e.g. 14.58 ± 2.54a). This should be changed.

• ''Our results showed in this study, ethanolic PPJE exhibited the highest antioxidant capacity'' - please indicate compared to what other extract?

•''It is worth noting that the extract produced with ethanol presented high antioxidant potential and also a high content of total phenols when compared with most of the extracts produced with organic solvents.'' This could have been due to the better solvation of antioxidant compounds present in this family as a result of interactions (hydrogen bonds) between the polar sites of the antioxidant molecules and the solvent. In conclusion, the cladode of O. stricta showed higher scavenging activity in ethanolic extract.'' No information was found in the submitted manuscript regarding the methodology or results of the different extraction methods performed with different solvents. So why this conclusion?

Author Response

Dear Editor,

Thank you for your email dated consideration of our manuscript entitled “HPLC analysis, antioxidant and the potential role of Opuntia stricta extract in preventing Cadmium-caused physiological disorders in rats: biochemical and molecular evidence". We wish to thank the editor and reviewers for their helpful comments. We have now thoroughly revised the manuscript by incorporating all changes suggested. The corrections are indicated by red colour in the revised version.

Reviewer#2 comments

Question 1: Fig 1. Should be corrected. The methodology does not indicate that this is an in vitro method but a basic laboratory analysis.

Response 1: We agree with the reviewer’s assessment. This figure has been changed.

Question2: In the abstract, statistical significance is reported at the 0.01 level, while in the statistical description, the confidence level was set at 0.05. Please verify this information.

Response 2: We agree with the reviewer’s assessment. This mistake has been checked.

Question 3: ''[26] found that Cd-exposure induced subcellular accumulation of hydrogen peroxide'', Please correct to by Name et al. [year].

Response 3: Correction is made

Question 4: In many cases, spaces are missing between words/characters in the manuscript

Response 4: Correction is made.

Question 5: The subsection ''Enzymatic antioxidant status'' is not numbered and is not in italics.

Response 5: In response to the reviewers’ suggestion. The mistake has been checked.

Question 6: The subject of this study was not cladode juice but cladode juice ethanol extract, please detail this information in the missing places.

Response 6: We agree with the reviewer’s assessment. The objective was determined the antioxidant and preventive actions of phenolic extract isolated from O. stricta cladode juice. The ethanol was used as solvent ton extract these phenolic copmpounds.

Question 7: Please correct were evaluated ''using using'' for evaluated using

Response 7: Correction is made.

Question 8: Conducting appropriate statistical analysis is critical to the research being conducted. What confidence levels in ''4.11. Statistical analysis'' were assigned to each analysis, please annotate and verify throughout the manuscript.

Response 8: We agree with the reviewer’s assessment. The confidence level has been added in 4.11 section.

Question 9: In Table 4,5,6, the abbreviation denoted in the superscript (''SOD a, CAT b, GPx c, LPO d, Proteine carbonyl e'') may not be the same as the denotation of statistical differences between the results presented (e.g. 14.58 ± 2.54a). This should be changed.

Response 9: The superscript has been changed by other abbreviations.

Question 10: ''Our results showed in this study, ethanolic PPJE exhibited the highest antioxidant capacity'' - please indicate compared to what other extract?

Response 10: We agree with the reviewer’s assessment. The sentence has been revised.  

Question 11: ''It is worth noting that the extract produced with ethanol presented high antioxidant potential and also a high content of total phenols when compared with most of the extracts produced with organic solvents.'' This could have been due to the better solvation of antioxidant compounds present in this family as a result of interactions (hydrogen bonds) between the polar sites of the antioxidant molecules and the solvent. In conclusion, the cladode of O. stricta showed higher scavenging activity in ethanolic extract.'' No information was found in the submitted manuscript regarding the methodology or results of the different extraction methods performed with different solvents. So why this conclusion?

Response 11: We agree with the reviewer’s assessment. The sentence has been removed.

Reviewer 3 Report

The current paper discusses the protective effects of Opuntia stricta juice extract against hepato-nephrotoxicity, testicular injury, and DNA damage induced by cadmium and also performed chemical profiling of extracts using HPLC.

Specific comments:

Title: Scientific names should be italicized.

Introduction: Authors are advised to check the citations throughout the introduction section. The description on medicinal plants only has one citation (related to ginger and Atorvastatin) and should be improved to emphasize the potential of natural products derived from (medicinal) plants.

Results:

Please improve this difficult-to-understand sentence: “DPPH percent scavenging activity was a determined different concentrations of juice extract. “

DL50, LD or LD50? Please check on consistencies.

Figure 4: Is there any DNA ladder?

Figure 7, 8, 9: Images stacked over one another. Can’t see properly.

Tables are difficult to read (i.e., overlapped, maybe due to pdf conversion?)

Methods:

Control mice received five weeks of normal saline for 5 weeks. Did Group 2-4 receive Opuntia stricta juice extract/Cd for five weeks too? Please clarify in the methods section.

Please check subheadings for methods section.

Discussion:

The authors are encouraged to revise the entire discussion section.

DL50, LD or LD50? Please check on consistencies.

Conclusion:

O. stricta juice extract present the highest values of phenolic and flavonoids content…”

What is the comparator extract? Please improve the sentence.

Author Response

Dear Editor,

Thank you for your email dated consideration of our manuscript entitled “HPLC analysis, antioxidant and the potential role of Opuntia stricta extract in preventing Cadmium-caused physiological disorders in rats: biochemical and molecular evidence". We wish to thank the editor and reviewers for their helpful comments. We have now thoroughly revised the manuscript by incorporating all changes suggested. The corrections are indicated by red colour in the revised version.

Reviewer#3 comments

Question 1: Title: Scientific names should be italicized.

Response 1: We agree with the reviewer’s assessment. Correction is made

Question2: Introduction: Authors are advised to check the citations throughout the introduction section. The description on medicinal plants only has one citation (related to ginger and Atorvastatin) and should be improved to emphasize the potential of natural products derived from (medicinal) plants.Response 2: We agree with the reviewer’s assessment. This sentence has been added in the introduction section :

In recent years, studies on antioxidant activities of medicinal plants have increased remarkably due to increased interest in their potential of being used as a rich and natural source of antioxidants. Many medicinal plants contain large amounts of antioxidants such as polyphenols, which can play an important role in adsorbing and neutralizing free radicals, quenching singlet and triplet oxygen, or decomposing peroxides. The beneficial health effects of plants are attributed to flavonoids, a class of secondary metabolithathich protect the plant against ultraviolet light and even herbivores.

Question 3: Please improve this difficult-to-understand sentence: “DPPH percent scavenging activity was a determined different concentrations of juice extract. “

Response 3: Correction is made

Question 4: DL50, LD or LD50? Please check on consistencies.

Response 4: This sentence has been removed.

Question 5: Figure 4: Is there any DNA ladder?

Response 5: In response to the reviewers’ suggestion. This is a preliminary test to test the effect of cadmium exposure on DNA fragmentation.

Question 6: Figure 7, 8, 9 : Images empilées les unes sur les autres. Impossible de voir correctement.

Response 6: We agree with the reviewer’s assessment. This mistake has been revised.

Question 7: Tables are difficult to read (i.e., overlapped, maybe due to pdf conversion?)

Response 7: We agree with the reviewer’s assessment. This mistake has been revised.

Question 8: Control mice received five weeks of normal saline for 5 weeks. Did Group 2-4 receive Opuntia stricta juice extract/Cd for five weeks too? Please clarify in the methods section

Response 8: We agree with the reviewer’s assessment. This sentence has been clarified.

Question 9: Please check subheadings for methods section.

Response 9: The subheadings have been checked.

Question 10: The authors are encouraged to revise the entire discussion section.

Response 10: We agree with the reviewer’s assessment. This section has been revised.  

Question 11: ' DL50, LD or LD50? Please check on consistencies.

Response 11: We agree with the reviewer’s assessment. The sentence has been removed.

Question 12: O. stricta juice extract present the highest values of phenolic and flavonoids content…”

What is the comparator extract? Please improve the sentence.

Response 12: We agree with the reviewer’s assessment. The sentence has been corrected.

Question 13: The image for gel electrophoresis remains the same (white patch and no ladder marker) and it's best to have ladder marker as (positive) control and "ruler" to estimate size of the nucleic acids for the experiment.

Response 13: We agree with the reviewer’s comment. I eliminated figure 3 (DNA damage) because I only have this photo without a size marker and this study was done in 2017 I do not have access to the samples. 

Question 14: Table 3: "Groups" not "groupes"
Table 4: Extra space for "Group 4"
It would be good to rename all the groups for tables/figures (e.g. Control,
Opuntia stricta extract, CdCl2-treated model, CdCl2-treated + Opuntia stricta
extract)."

Response 14:  We agree with the reviewer’s comment. All corrections are made.